# Muscle Function, Body Composition, Insulin Sensitivity and Physical Activity in Adolescents Born Preterm: Impact of Gestation and Vitamin D Status

**DOI:** 10.3390/nu14235045

**Published:** 2022-11-27

**Authors:** Claire L. Wood, Robert Tinnion, Kieren G. Hollingsworth, Michael I. Trenell, Mark S. Pearce, Tim D. Cheetham, Nicholas D. Embleton

**Affiliations:** 1Department of Paediatric Endocrinology, Royal Victoria Infirmary, Newcastle upon Tyne NE1 4LP, UK; 2Translational and Clinical Research Institute, Faculty of Medical Sciences, Newcastle University, Newcastle upon Tyne NE2 4HH, UK; 3Newcastle Neonatal Service, Royal Victoria Infirmary, Newcastle Hospitals NHS Foundation Trust, Newcastle upon Tyne NE1 4LP, UK; 4Population Health Sciences Institute, Faculty of Medical Sciences Newcastle University, Newcastle upon Tyne NE1 4LP, UK

**Keywords:** preterm, muscle, vitamin D

## Abstract

Whilst several studies have explored adolescent metabolic and cognitive function after preterm birth, few have explored muscle function and physical activity. We set out to examine the relationship between gestational age and muscle metabolism in a cohort of adolescents who were born preterm. Participants were recruited from the Newcastle preterm birth growth study cohort. They did not have severe neurological disease and were not on daily medication. Participants underwent an assessment of oxidative muscle function using phosphorus magnetic resonance spectroscopy that included the half-time for recovery of equilibrium of phosphocreatine, τ_½_PCr. In addition, we measured key variables that might affect muscle function including physical activity levels determined by 3-day accelerometry, body composition using air displacement plethysmography, insulin sensitivity using the homeostatic model assessment/Matsuda index and serum vitamin D concentrations. 60 adolescents (35F) median age 15.6 years (range 12.1–18.8) with a median gestation of 31 weeks (range 24 to 34 weeks) underwent a single assessment. Males were more active and spent less time in sedentary mode. Time spent in light activity was associated with insulin sensitivity (IS) (Matsuda Index; *p* < 0.05) but there were no strong correlations between activity levels and gestational age. Greater fat mass, waist circumference and body mass index were all associated with lower IS. Gestational age was negatively associated with adjusted measures of oxidative muscle function (τ_½_PCr). In a stepwise multivariate linear regression model, gestational age at birth was the most significant predictor of oxidative muscle function (*p* = 0.005). Higher serum vitamin D levels were also associated with faster phosphocreatine recovery time (*p* = 0.045). Oxidative function in the skeletal muscle of adolescents born preterm is associated with gestational age and vitamin D concentrations. Our study suggests that being born preterm may have a long-term impact on muscle metabolism.

## 1. Introduction

Preterm birth is associated with an increased risk of adverse outcomes in later life [1] with higher rates of obesity, type 2 diabetes (T2DM), cardiovascular disease, and other features of the metabolic syndrome [2]. The mechanisms are likely to be complex and may include epigenetic effects as well as effects on organogenesis with altered tissue development [1]. Prematurity may also impact on skeletal muscle development and function, which is important because skeletal muscle oxidative function is a key component of metabolic health [3]. Preterm delivery results in less time spent in the muscle-conditioning intra-uterine environment and can also be associated with lower rates of physical exercise in later life [4]. Insulin resistance in adults with T2DM is more common following preterm birth and limits the availability of glucose to the mitochondria with studies using phosphorus magnetic resonance spectroscopy (^31^P-MRS) showing this can compromise skeletal muscle function [5]. Gentle regular exercise does not appear to improve substrate delivery in adults with T2DM but enhances insulin sensitivity by increasing lipolysis in skeletal muscle [6]. Changes in organogenesis, conditioning and subsequent use of skeletal muscle following preterm birth may therefore be important determinants of metabolic health outcomes. This is important because targeted exercise can reduce obesity in teenagers including those born preterm and offers potential strategies for reversing or ameliorating the adverse effects of preterm birth.

^31^P-MRS is a non-invasive technique for assessment of organ structure and metabolite concentrations. The concentration of phosphorus nuclei (^31^P) in chemically distinct compounds within muscle, including high energy metabolites such as PCr and ATP, can be quantified using a clinical MRI scanner with modified hardware. The use of a graded exercise apparatus to perform plantar flexion exercises can be used to deplete PCr stores while ATP homeostasis is preserved. Cessation of exercise permits oxidative phosphorylation of ATP and recovery of PCr concentrations. The rate of recovery to baseline PCr concentrations within muscle (denoted as τ_½_PCr) is used as a measure of mitochondrial oxidative function. 

Vitamin D is as steroid hormone partly acquired through dietary sources but mainly generated by UV light passing through the skin and converting 7-dehydrocholesterol to vitamin D3. This then undergoes hydroxylation in the liver and then kidney to produce 1,25 dihydroxyvitamin D3. Effects of vitamin D are mediated throughout the body by vitamin D receptors in both bone and non-skeletal tissues such as muscle. Vitamin D deficiency in adults has been shown to have an impact on mitochondrial oxidative function when measured using the ^31^P-MRS technique [7]. As vitamin D is primarily transferred to the fetus in the third trimester, premature infants are at high risk of vitamin D insufficiency and deficiency. Studies from the UK and Ireland have shown that at least two thirds of premature infants are likely to have vitamin D deficiency at birth [8,9] In addition, current neonatal nutritional strategies for early preterm infants may be insufficient to achieve recommended vitamin D intake and target serum 25(OH)D concentrations [8].

Serum vitamin D levels correlate with insulin sensitivity and vitamin D supplementation improves peripheral muscle insulin sensitivity, therefore it is important to consider vitamin D levels when investigating the association between preterm birth, insulin sensitivity and muscle function. 

Few studies have explored the impact of preterm delivery on muscle metabolism and function in later life. We aimed to examine the association between gestational age and muscle function and metabolism using ^31^P-MRS in late adolescence after adjustment for likely confounders including insulin sensitivity, fat mass, vitamin D status and physical activity levels. 

## 2. Materials and Methods

### 2.1. Participants and Visit Schedule

This study was conducted in adolescents who were recruited from the Newcastle preterm birth growth study (PTBGS) cohort [10]. Participants were preterm infants (gestation range 24–34 weeks and mean birthweight ~1.2 kg) originally enrolled into one of two randomised neonatal nutrition trials [11,12]. Participants were excluded from the original studies if they had major neonatal morbidities such as severe neurological disease, chronic lung disease at discharge or were on regular daily medication. Of the 247 children in the PTBGS cohort, details were obtained for 235 using health care records, following confirmation from the general practitioner that children were alive and otherwise well. Families were contacted by letter, with one further follow up letter sent if no reply was received. The parents of 62 adolescents/adolescents themselves responded, of whom 60 attended for a half-day study visit at the Newcastle Magnetic Resonance Centre, Newcastle University Campus for Ageing and Vitality, UK. Study visits were conducted following an overnight (minimum 8-h) fast. Participants had the option of declining any of the study components. The protocol for the study has previously been published [10].

### 2.2. Anthropometry including Body Composition Determination

Participants underwent standard anthropometry (height, weight, waist circumference) by an appropriately trained clinician/researcher (RT). Each measurement was taken three times and averaged. Body mass index (BMI) was calculated (weight in kg divided by height in metres squared). Total body fat mass (FM) was measured by Air-Displacement Plethysmography (BOD POD™ system, COSMED, Concord California, USA) [13]. Tight-fitting swimsuits and a swim-cap were worn in the BODPOD to minimise clothing/hair volume interference with the measurement. Thoracic gas volume was not directly measured; instead a standardised, software-generated estimate was used to minimise the length of time in the BODPOD [14]. Standard deviation z-scores were calculated for birthweight, height, weight and BMI using reference data [15,16]. Self-assessed pubertal (Tanner) stage was recorded [17]. 

### 2.3. Biochemistry including Assessments of Insulin Sensitivity and Vitamin D Status

Oral glucose tolerance testing [18] was carried out and both Homeostatic model assessment (HOMA) modelling [19] and Matsuda Indexing [20] were used to assess IS based on T_0min_ and T_120min_ serum values. HOMA is a marker of hepatic IS and requires only a baseline glucose and insulin concentration. The Matsuda index is a measure of both hepatic and peripheral IS and is calculated following a glucose load. Vitamin D concentrations were measured because of the potential association with fat mass and oxidative muscle function [21,22]. Serum was stored frozen prior to assay for 25-OH Vitamin D quantitation using an ABSciex 5500 tandem mass spectrometer (Warrington, UK) and the Chromsystems (Munich, Germany) 25OHD kit for LC-MS/MS (intra- and inter-assay CV 3.7% and 4.8%, respectively) at Manchester University. Triglycerides were measured using a colorimetric method (Roche Diagnostics GmbH, Mannheim, Germany) with inter assay CV’s of 0.9–2.0%.

### 2.4. Assessment of Physical Activity Levels

Study participants wore a lightweight, uniaxial accelerometer (ActiGraph™ Pensacola, Florida, USA) over 3 days following a study visit, which was pre-programmed to begin and stop recording activity prior to being issued to participants. They measured acceleration of different intensities in one (vertical) plane during sequential, pre-defined, short epochs (10 s). During wearing, any displacement of the hip (i.e., pelvis tilt or changing body position) during activity causes vertical displacement of the accelerometer which is recorded as counts per minute (cpm) [23]. Analysis of the activity data were performed using the Actilife program (version 5, MTI, Pensacola, Florida, USA) under the supervision of Dr Laura Basterfield (Human Nutrition Research Centre, Newcastle University). Using pre-defined, calorimetrically referenced cut-offs, the data recorded over three consecutive days was classified into sedentary (≤100 cpm), light (101–2295 cpm), moderate (2296–4011 cpm) or vigorous (≥4012 cpm) physical activity (MVPA) [24]. 

### 2.5. Magnetic Resonance Spectroscopy

A 3-Tesla Achieva MRI scanner (Philips) was used with a 14 cm ^31^P surface coil placed over the calf muscles (Figure 1B) to acquire phosphorus spectra [25]. ^31^P spectra were recorded over 3 min to assess resting mitochondrial function, then participants performed plantar flexion against resistance (at 35% MVC loading, see Appendix A) at 0.5 Hz for 3 min. Participants then rested and MR phosphorous spectra were recorded every 10 s. ^31^P spectra were analysed using the AMARES algorithm in jMRUI and prior knowledge [26]. From this, quantification of PCr, inorganic phosphate (Pi) and pH was obtained (Figure 1A). A single exponential fit was used to estimate half-time to recovery within the muscle (Figure 1D), represented by recovery of equilibrium of PCr (τ_½_PCr) and ADP (τ_½_ADP). 

### 2.6. Statistics

The precise sample sized was linked to the number of respondents but according to effect size data from the cohort at earlier assessments where fat mass index was related to insulin sensitivity (R^2^ = 0.12), 60 patients were sufficient to detect a within-cohort difference with 80% power at the 5% significance level. However, the analyses in this paper included more powerful linear regression methods, using mainly continuous data suggesting that the numbers involved in this study were adequate. Stata™ (v11/13 StataCorp, College Station, TX, USA) was used for all statistical analyses. Spearman correlation analysis was used to examine potential associations between paired variables. Mann–Whitney U or ‘t’ tests indicated that the follow-up sample was broadly representative of the original cohort for key variables including gestational age, birthweight and birth weight SDS. No active selection process was used when contacting families (i.e., no discriminators based on previous cohort results) with all respondents included in the study. We are not, therefore, aware of any recruitment bias. Multiple linear regression modelling was used to examine interactions between variables using τ_½_PCr as the dependent variable. Key variables that could impact on muscle function were considered when comparing gestational age to muscle metabolism. Sex, birthweight, activity level, FMI, IS and vitamin D concentration were all analysed separately in univariate models, then a manual forwards stepwise approach used to inform the final variables for inclusion into the multivariable regression model. Non-significant terms were removed until the minimum number of significant variables remained. Age at follow-up was not included as it was felt that including both pubertal stage and age in a model may lead to collinearity (Rho = 0.62, *p* < 0.001). Indices of insulin sensitivity were logarithmically transformed as residuals deviated from homogeneity. Statistical significance was set at 5%. 

## 3. Results

### 3.1. Baseline Characteristics and Recruitment Flowchart

60 adolescents (15.6 years old, range 12.1–18.8) attended a study visit, although not all agreed to each component (Table 1). Participants were representative of the original Newcastle PTBGS cohort and there was no difference between those consenting to each component and the overall cohort in terms of age at study, gender, gestation at birth, birthweight or body fat content recorded at previous visit. 

### 3.2. Anthropometry 

The height and weight of participants is shown in Table 1. Most (89%) were in mid to late puberty (Tanner stage 3–5) at time of assessment. Body composition data were consistent with UK age and sex-matched reference data. Median FMI for males was 1.93 (9th–25th centile) and LMI 16.55 (50th centile) and females FMI 5.45 (25th–50th centile) and LMI 15.70 (50th–75th centile). Markers of adiposity were also analysed with respect to the measured muscle kinetics and no significant correlation identified. 

### 3.3. Insulin Sensitivity

Measures of IS using HOMA and Matsuda index are shown in Table 2. 

Body composition and greater fat mass, waist circumference, and higher BMI z-score were all associated with lower IS, but there was no correlation between gestational age or birthweight z-score and IS (Table 3). Spearman correlation analysis of recorded minimum pH during standardised exercise showed no correlation between minimum pH and IS measured by Matsuda (Rho:0.07, *p* = 0.75). Serum triglyceride levels were also analysed with respect to measured muscle kinetics and no significant correlation identified. 

### 3.4. Accelerometery 

Valid accelerometery results (>6 h/day wear-time) were obtained in 44/60 participants. Mean daily time spent in MVPA in the overall cohort was 45 min, 25% less than the nationally recommended 60 min per day [27,28]. Average step count was also less than the recommended 10,000 daily steps [29]. Males were more active than females (Table 2) with a significantly higher proportion of their time spent in light activity (*p* = 0.02) and a lower proportion in sedentary mode (*p* = 0.02), but there was no significant difference in the time spent in MVPA. There were no strong correlations between activity levels and gestational age or markers of adiposity. Time spent in light activity was significantly associated with the Matsuda Index of IS (Table 3). There was no correlation between accelerometer-measured activity levels and muscle metabolism, and no measurable effect of sex (Table 4).

### 3.5. Vitamin D (25-OHD) Concentrations 

The median serum Vitamin D concentration was 59.3 nmol/L (range 21.9–102.6 nmol/L; Table 1) and was deficient/insufficient in 12/46 (26%) participants. Levels were significantly higher in visits occurring between March-September compared to October-February (*p* = 0.001). Spearman correlation analysis showed a negative association between vitamin D levels and BMI (Rho: −0.31, *p* = 0.046) but not fat mass (Rho: −0.27, *p* = 0.07) or waist circumference (Rho: −0.28, *p* = 0.06). Vitamin D was positively associated with enhanced IS when measured by Matsuda index (Rho: 0.41, *p* = 0.005) and HOMA (Rho: 0.34, *p* = 0.02). The greater the vitamin D level, the shorter the half-time to recovery of equilibrium of PCr (co-eff-0.11, *p* = 0.03).

### 3.6. Magnetic Resonance Spectroscopy

Magnetic resonance spectroscopy was performed in 50 participants. Spearman rank correlation analysis and univariate linear regression modelling were used to determine independent factors associated with oxidative muscle function (measured by τ_½_PCr). Gestational age was negatively associated with τ_½_PCr, i.e., the greater the gestational age at birth, the shorter the half-time to recovery (Table 4). Birthweight z-score was not associated with MRS kinetics.

### 3.7. Stepwise Linear Regression with τ_½_PCr as the Outcome Variable

Multivariable linear regression modelling was undertaken, with τ_½_PCr as the outcome variable. Because IS and activity levels were associated with vitamin D levels, these parameters were tested for inclusion in the initial model. As anticipated, IS was related to puberty stage: Matsuda Index was significantly higher in those who were pre-pubertal (Tanner stage 1) compared to those who were Tanner stage 2–5; 8.83 v 5.16 (coefficient 1.21 (0.14, 2.25), *p* = 0.03). Pubertal staging was therefore considered in the stepwise model. Variables tested for inclusion within the final model are in Table 4. In the final regression model (data from 36 participants with MRS data and vitamin D levels recorded) gestational age at birth remained the most significant predictive factor of oxidative muscle function; those born more premature had longer phosphocreatine recovery times (co-efficient −0.26, *p* = 0.005). Higher vitamin D levels were also associated with faster recovery times (co-efficient—5.86, *p* = 0.045, *p*-value for overall model = 0.004, adjusted R^2^ = 0.24. Testing for heteroskedasticity showed that the final terms (vitamin D and gestational age) were independent. Inclusion of Tanner Staging was not significant by likelihood ratio testing of this model with/without Tanner Staging included.

## 4. Discussion

We have shown that gestational age at birth is predictive of skeletal muscle oxidative function in later life as measured by τ_½_PCr. This demonstrates the life-long adverse metabolic consequences of preterm birth. We detected the anticipated associations between puberty stage, BMI, fat mass and activity levels with IS but, unlike gestational age, none of these variables predicted skeletal muscle oxidative function. We found that lower vitamin D concentrations were associated with worse skeletal muscle oxidative function as has been described previously [20].

There are few similar studies. Work by Bertocci showed that infants who were born preterm have a smaller PCr signal than their term born counterparts, using a similar MRS technique to that used here [30]. They speculated that preterm infants may have limited phosphate reserves such that when challenged by even a small increase in activity they deplete their reserves very quickly and take a long time to recover. However, this mechanism would not explain persistence of this effect in later life. Using other methods to measure oxidative capacity (aerobic fitness score), Rogers and colleagues demonstrated that at age 17, adolescents born preterm have reduced oxidative capacity and are more likely to become fatigued [31].

We believe that the association that we have identified is biologically plausible because post-mortem specimens from the skeletal muscle of infants born preterm show a predominance of type 2 (low-oxidative) muscle fibres which have a relatively low density of mitochondria [32,33,34]. In these studies, which looked at muscle groups such as the diaphragm and intercostal muscles, data suggest that typical development towards term age involved an increase in the density of type 1 (high oxidative) fibres. Some preterm infants may therefore experience an arrest or permanent alteration in the density of type 1 fibres associated with lower mitochondrial density that may then result in a prolonged PCr recovery in later life. Habitual activity levels might be associated with the oxidative capacity of the mitochondria if it was modifiable by muscle training, but we found no evidence of a correlation between an objective measurement of daily activity and ^31^P-MRS spectroscopy. However, absence of such an association may reflect the fact that the test standardised to MVC is designed to deplete the PCr stores substantially to measure oxidative function during recovery of PCr, and thus is perhaps more extreme than the more usual low-level of aerobic exercise.

We noted a strong association between vitamin D concentrations and muscle kinetics which is important as it is potentially modifiable [21]. The relationship between lower serum vitamin D levels and longer τ_½_PCr recovery time was independent of the relationship between τ_½_PCr and gestational age at birth in our cohort. Vitamin D concentrations tend to be lower in obese individuals and have been shown to be lower at birth in babies born preterm [35]. In other populations there is also a well-documented association between lower vitamin D levels and markers of reduced IS [36,37,38,39]. Other data show that vitamin D replacement improves IS [40,41]. Both FM and IS were therefore considered for inclusion in the multivariable linear regression but were not significant variables in the final model. 

Vitamin D has a wide range of effects in tissues and cells, but there are no definitive data to show that it has a direct effect on muscle at the myocyte membrane. It is possible that it may act at the mitochondrial genomic level upregulating either the cellular mechanisms needed to improve oxidative function, or more directly by influencing ATP synthesis [42]. Further studies could explore the potential for Vitamin D supplementation to improve muscle function in this group with a particular focus on patients with vitamin D levels at the lower end of the spectrum observed in this study. 

Obtaining valid accelerometery data in the adolescent population is challenging and only 44/60 completed >6 h wear-time. The directly measured activity data from this cohort was similar to another cohort in our region which also noted that boys were more physically active than girls [24,43,44]. Another striking feature from our data was that light activity is most strongly associated with measures of IS. While there was no direct association between τ_½_PCr and any of the activity measures, increasing light activity may have an indirect effect on muscle oxidative function via either its positive association with IS or via mechanisms that link IS with vitamin D. The association of body habitus with IS across the cohort longitudinally [45] is in agreement with other published data describing the increasing influence of body habitus on IS with increasing chronological age [46]. 

The association between light physical activity and improved glucose disposal reflects the relationship between exercise and IS in muscle that has been observed in a range of different settings. Light activity appears to have the most impact as observed in animal models [47], older humans [48] and individuals at high risk of metabolic syndrome [49], and promotion of this may be more acceptable in adolescents than attempts to increase more vigorous forms of activity. 

### Limitations

This study has important limitations. The cohort is modest in size, however the group studied appeared to be representative of the initial cohort with respect to key characteristics. Our sample size was fixed by virtue of the size of the initial cohort and then the number of willing respondents, but the study was sufficiently powered to detect a change in τ_½_PCr of 10%. It is of note that the overall τ_½_PCr values are not indicative of major defects in oxidative function, as might be found in mitochondrial dysfunction, and indeed are similar to a group of healthy young adults also tested under the same protocol [50]. The differences may be relevant, however, in terms of an individual’s ability to participate competitively in sporting activities and so should not be trivialised. Whilst the association between gestational age and muscle metabolism in premature infants is not dependent on term controls, future studies need to assess muscle function in the full spectrum of gestational age. We also cannot rule out the fact that the study may have been underpowered for the non-significant outcomes and similarly the sample size was too small to stratify by gender. 

Assessing pubertal status in studies involving adolescents can be challenging although a recent meta-analysis concluded that there is moderate or substantial agreement between patient and clinician assessments [17]. The value of a clinician’s assessment of pubertal stage needs to be balanced against the likelihood of a n associated reduction in participant numbers. Importantly, self-assessed pubertal stage was related to IS and the relevance of pubertal status in this study is primarily because of measured variables such as fat-mass and fat-free mass. Reassessment of metabolic outcomes including muscle function in early adulthood is important.

## 5. Conclusions

We noted a relationship between gestational age and muscle oxidative capacity determined using ^31^P-MRS in adolescent children born preterm. This may be programmed by differential development of muscle fibre type as a result of preterm birth. The relationship between gestational age and muscle function in later life requires further investigation in a larger study but in the interim we recommend optimising vitamin D status and encouraging preterm-born children and adolescents to engage in regular light activity.

## Figures and Tables

**Figure 1 nutrients-14-05045-f001:**
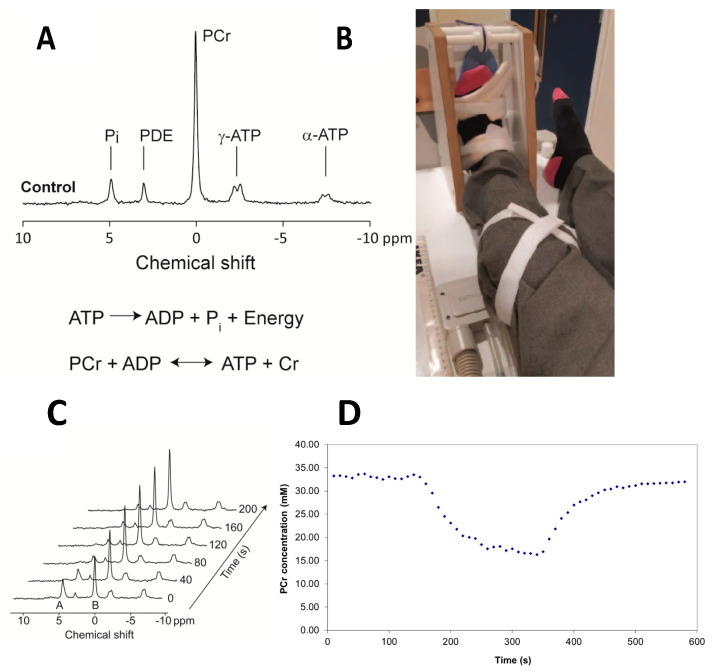
(**A**) The phosphorus spectrum of the human gastrocnemius/soleus muscle at rest, showing the resonances due to inorganic phosphate (Pi), phosphodiesters (PDE), phosphocreatine (PCr) and the α and γ phosphorus nuclei of ATP. The area under each peak indicates the concentration of that substance. (**B**) The apparatus used for the plantar flexion exercise showing the pedal, and the radiofrequency coil used to acquire signals from phosphorus-containing high energy compounds from the back of the leg during exercise and recovery. The pedal is attached to a weight within the scanner bore to provide a fixed resistance of 35% of the maximum voluntary contraction of the subject. (**C**) A dynamic series of phosphorus spectra acquired after the plantar flexion exercise has ceased, showing the changes in the inorganic phosphate peak and the phosphocreatine peak, which recovers to equilibrium. (**D**) Extracting the area under the curve of the phosphocreatine peak for each time point as concentration, showing the initial concentration at rest, depletion during the exercise and recovery by oxidative phosphorylation after exercise has ceased. The τ_½_PCr is calculated by fitting a monoexponential curve to the recovery. In this case, the τ_½_PCr was 33.2 s.

**Table 1 nutrients-14-05045-t001:** Subject characteristics at birth and time of follow-up and auxology, body composition and biochemistry at study visit. BMI: body mass index, SD: standard deviation.

Variable	Mean (SD) or Median (Range)
Sex (Male: Female)	25 (42%): 35 (58%)
Birth weight (kg), n = 60	1.37 (0.25)
Birthweight z-score, n = 60	−0.72 (−3.81, 1.37)
Gestation at birth (weeks), n = 60	31.0 (1.9)
Age (years), n = 60	15.6 (12.1–18.8)
Weight z-score, n = 60	0.02 (1.21)
Height z-score, n = 60	−0.19 (0.90)
BMI z-score, n = 60	0.16 (1.21)
Pubertal status by Tanner staging (1–5), n = 60	Stage 1—8%
	Stage 2—3%
	Stage 3—27%
	Stage 4—40%
	Stage 5—22%
25(OH)D (nmol/L), n = 46	59.3 (21.9–102.6)
Systolic blood pressure (mm Hg), n = 60	116.1 (13.0)
Diastolic blood pressure (mm Hg), n = 60	73.1 (7.24)
Mean blood pressure (mm Hg), n = 60	83.5 (7.5)
Cholesterol (mmol/L), n = 46	4.1 (3.2–6.5)
Triglyceride (mmol/L), n = 46	0.7 (0.4–2.6)
Total body fat mass from BodPod (kg/m^2^), n = 58	3.4 (0.4–15.1)
Total body fat-free mass from BodPod (kg/m^2^), n = 58	15.9 (13.2–20.1)
Waist circumference (cm), n = 60	71.7 (57.0–108.7)

**Table 2 nutrients-14-05045-t002:** Exercise and muscle function parameters assessed by accelerometery, oral glucose tolerance test and magnetic resonance spectroscopy. HOMA: Homeostatic model assessment, PCr: phosphocreatine.

Variable	Mean (SD) or Median (IQR)	Male	Female	*p*-Value for Male v Female
Moderate to vigorous physical activity (mins/day) n = 44	45.0 (22.8)	52.1 (21.5)	39.7 (22.7)	0.06
Sedentary activity (mins/day), n = 44	500.0 (84.2)	483.1 (93.2)	512.9 (76.0)	0.02
Light activity (mins/day), n = 44	143.3 (47.5)	154.7 (44.9)	134.7 (48.5)	0.02
Insulin sensitivity assessed by Matsuda index, n = 50	4.9 (3.0–7.1)	5.6 (3.5–8.7)	4.0 (2.9–5.2)	0.13
Insulin sensitivity assessed by HOMA (%), n = 46	79.7 (65.9–114.1)	98.1 (66.9–118.4)	78.3 (62.6–111.0)	0.32
PCr recovery time, τ_½_PCr (s), n = 50	33.8 (7.3)	31.5 (7.9)	34.2 (8.8)	0.28

**Table 3 nutrients-14-05045-t003:** Spearman’s correlation analysis for. (**A**) measures of physical activity, assessed by accelerometry. (**B**) insulin sensitivity (* *p* < 0.05, ** *p* < 0.01). BMI: body mass index, FMI: fat mass index, MVPA: moderate-vigorous physical activity.

**(A)**			
**Variable**	**MVPA (Mean Mins)**	**Sedentary Activity (Mean Mins)**	**Light Activity (Mean Mins)**
Gestational age	−0.26	0.02	0.04
FMI (Bodpod)	−0.18	0.03	−0.21
Current BMI z-score	0.00	0.11	−0.14
Waist circumference (cm)	−0.15	0.12	−0.25
Matsuda Index	0.03	0.05	0.32 *
**(B)**			
**Variable**	**Matsuda Index**	**HOMA %S**	
Gestational age	−0.05	−0.06	
Birthweight z-score	−0.03	−0.16	
FMI (Bodpod)	−0.41 **	−0.39 **	
Waist circumference (cm)	−0.30 *	−0.34 *	
BMI z-score	−0.41 **	−0.39 **	

**Table 4 nutrients-14-05045-t004:** Univariable regression analysis for adjusted PCr recovery (τ_½_PCr), showing the unadjusted values for the variables that were considered for inclusion in the multivariable model. FMI: fat mass index, * indicate variables that were included in final model, + indicates log transformation.

Independent Variable	Co-eff	95% CI	*p*-Value
Gestational age at birth (days)	−0.21	−0.39, −0.03	0.03 *
Serum Vitamin D +	−0.11	−0.21, −0.01	0.03 *
Fat mass index (measured by BodPod)	−1.07	−4.36, 2.21	0.52
Mean step-count/day	1.70	−3.86, 7.26	0.54
Matsuda index of insulin sensitivity+	3.01	−1.32, 7.35	0.17
Birthweight z-score	1.15	−1.46, 3.75	0.38
Female sex	2.61	−2.23, 7.45	0.28

## Data Availability

Data are available on request.

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
