# Peer review of "Muscle Function, Body Composition, Insulin Sensitivity and Physical Activity in Adolescents Born Preterm: Impact of Gestation and Vitamin D Status"

_nutrients, 2022, doi:10.3390/nu14235045_

Round 1

Reviewer 1 Report

In the current manuscript, Wood et al assessed the impact of gestation and vitamin D status on muscle function, body composition, insulin sensitivity and physical activity in adolescents born preterm. The research is interesting and scientific valuable, but I also have several concerns listed below.

1.      The major concern is the sample size, even the authors claimed 60 participants is enough power, but the n in Table 2 is below 50, which may not good for further result interpretation. Moreover, the physical characteristics are different between male and female adolescents, the gender-stratified assessment need a lager sample size.

2.      The brief introduction did not contain enough information for readers unfamiliar with the field of the current study. For example, even the name of vitamin D occurred in the title, but there was no occurrence in the introduction. I recommend the authors rewrite the introduction based on the logical structure of the current study, make the manuscript easier to follow.

3.      How did the authors select covariates and which covariates were included in the regression models?

4.      Line 145, the ethical statement should not be put in the section of statistical method.

5.      Table 1. Since the author used “Male: Female” in the first row, the second column name should be modified.

6.      Table 1. The presentation of pubertal status looks confusing.

7.      Line 159. Is there Table 1 or Table I?

8.      Line 235. There is no Table V in the manuscript, if it is in the supplemental material, please modify it to manuscript.

Author Response

  1. The major concern is the sample size, even the authors claimed 60 participants is enough power, but the n in Table 2 is below 50, which may not good for further result interpretation. Moreover, the physical characteristics are different between male and female adolescents, the gender-stratified assessment need a lager sample size.

Thank you for your comments. Whilst we feel that the sample was modest rather than small, we would like to point out that the numbers were linked (inevitably) to the number of subjects from the original cohort prepared to take part. We were – importantly – satisfied from the outset that our study numbers had the power to detect a change in t1/2 PCr of 10% (see lines 158-159), which was the primary outcome. The Newcastle Preterm Birth Growth Study  represents a unique cohort containing a high level of detail of early nutritional and growth exposures, and a comprehensive assessment over time. To the best of our knowledge, there is no other similar cohort with this level of information available.

  1. The brief introduction did not contain enough information for readers unfamiliar with the field of the current study. For example, even the name of vitamin D occurred in the title, but there was no occurrence in the introduction. I recommend the authors rewrite the introduction based on the logical structure of the current study, make the manuscript easier to follow.

Thank you for your suggestion. We have included additional information into the introduction and structured it in the order that the results are reviewed, see above. We hope that this makes it clearer to understand.

  1. How did the authors select covariates and which covariates were included in the regression models?

We have amended the section describing how covariates were included in the regression models, to hopefully make it clearer. We have stated on lines 171-77 “Sex, birthweight, activity level, FMI, IS and vitamin D concentration were all analysed separately in univariate models, then a manual forwards stepwise approach used to inform the final variables for inclusion into the multivariable regression model, non-significant terms were removed until the minimum number of significant variables remained.  “We have added in an additional sentence “Age at follow-up was not included as it was felt that including both pubertal stage and age in a model may lead to collinearity”, which hopefully makes the inclusion process clearer for the reader to understand. 

  1. Line 145, the ethical statement should not be put in the section of statistical method.

Thank you, we have moved the ethical statement as requested.

  1. Table 1. Since the author used “Male: Female” in the first row, the second column name should be modified.

Thank you, we have modified the first row name to help make this clearer. Please confirm we have understood this comment correctly.

  1. Table 1. The presentation of pubertal status looks confusing.

Sorry, we apologise for this- the formatting has not transferred across properly when the document has uploaded. We have amended this and hopefully it looks better now.

  1. Line 159. Is there Table 1 or Table I?

Sorry for the typo- Table I has been amended to table 1

  1. Line 235. There is no Table V in the manuscript, if it is in the supplemental material, please modify it to manuscript.

Apologies- we merged the data from two tables to one during the final read through of the manuscript and therefore this should read “Table 4”

Reviewer 2 Report

I express my gratitude for the opportunity to review this manuscript for possible publication.
The manuscript tries to address a very interesting and current topic regarding body composition and physical activity in adolescents born preterm. However, there are some points whose clarity and interpretation may not be the most adequate or subject to reformulation.

We therefore hope to help authors to improve the structure and content of the manuscript.
Although the content is well argumented, I would invite the authors to focus attention on the following:

Introduction:

The theoretical framework is scarce, they should clearly describe the scientific evidence that supports the hypothesis they have raised. I suggest to describe in exhaustive way the relationship between muscle function, body composition, insulin sensitivity and physical activity in adolescents born preterm. Moreover, authors should clearly describe the scientific evidence that supports the importance of gestation and vitamin D status.

Methods section:

-        Experimental procedures should be better defined

-        More information should be provided about the participants’ characteristics.

-        Anthropometric measurements and physical tests presuppose a protocol. This element is missing from the methodological description, which may imply an impossibility of replicating the study due to a lack of clarity in this regard.

1.     I would like to see more of the practical implications. Based on the analyzed variables, how the authors intend to use their findings?

2.     Include DOI in all REFERENCES

Kind regards

Round 2

Reviewer 1 Report

In this version of manuscript named “Muscle function, body composition, insulin sensitivity and physical activity in adolescents born preterm: impact of gestation and vitamin D status”, the authors made great effort to improve the article. I am mostly satisfied with the revising, but still have two moderate comments.

1.      Firstly, the authors claimed this study has enough sample size and statistical power, but didn’t response my concern. I suggest the authors modified their expression “The precise sample sized was linked to the number of respondents but according to effect size data from the cohort at earlier assessments where fat mass index was related to insulin sensitivity (R2=0.12), 60 patients were sufficient to detect a within-cohort difference with 80% power at the 5% significance level. Stata™ (v11/13 StataCorp, Texas, USA) was used for all statistical analyses.”, and recalculate this part with real sample size used in statistical models. If the real sample size cannot support enough power to original statistical method, other methods should under consideration.

2.      Moreover, the sample size limited conducting gender-stratified assessment should be added in limitation section.

3.      The authors used a manual forward stepwise approach to select final covariates, please add the threshold of the stepwise approach. And, a correlation test result should be added to support the authors’ concern on collinearity.

Author Response

In this version of manuscript named “Muscle function, body composition, insulin sensitivity and physical activity in adolescents born preterm: impact of gestation and vitamin D status”, the authors made great effort to improve the article. I am mostly satisfied with the revising, but still have two moderate comments.

  1. Firstly, the authors claimed this study has enough sample size and statistical power, but didn’t response my concern. I suggest the authors modified their expression “The precise sample sized was linked to the number of respondents but according to effect size data from the cohort at earlier assessments where fat mass index was related to insulin sensitivity (R2=0.12), 60 patients were sufficient to detect a within-cohort difference with 80% power at the 5% significance level. Stata™ (v11/13 StataCorp, Texas, USA) was used for all statistical analyses.”, and recalculate this part with real sample size used in statistical models. If the real sample size cannot support enough power to original statistical method, other methods should under consideration.

Thank you for raising this important point. We have removed the initial sentence regarding power as we felt it was misleading and have added in further information into the methods section to inform the reader regarding power.  “However, the analyses in this paper included more powerful linear regression methods, using mainly continuous data suggesting that the numbers involved in this study were adequate.” A traditional retrospective power calculation would be very difficult in this scenario as multivariate linear regression modelling and multiple outcomes were used. As there were significant findings in the study, however, we can be reassured that there was indeed adequate statistical power.

  1. Moreover, the sample size limited conducting gender-stratified assessment should be added in limitation section.

Thank you- we have added this into the limitations section- “ We also cannot rule out the fact that the study may have been underpowered for the non-significant outcomes and similarly the sample size was too small to stratify by gender.”

  1. The authors used a manual forward stepwise approach to select final covariates, please add the threshold of the stepwise approach. And, a correlation test result should be added to support the authors’ concern on collinearity.

Thank you- we have added the p-value threshold and the correlation test result (highlighted in red in the revised paper)